# Biostimulant Application Enhances Fruit Setting in Eggplant—An Insight into the Biology of Flowering

**Alicja Pohl, Aneta Grabowska, Andrzej Kalisz**  **and Agnieszka Sękara ***

Department of Vegetable and Medicinal Plants, University of Agriculture in Krakow, 29-Listopada 54,
31-425 Kraków, Poland
* Correspondence: agnieszka.sekara@urk.edu.pl; Tel.: +48-12-662-52-16

**Abstract:** Eggplant (*Solanum melongena* L.) is a warm climate crop. Its cultivation extends to temperate regions where low temperatures can affect the course of the generative phase, which is primarily sensitive to abiotic stress. The novelty of the present investigation consisted of characterising the heterostyly, pollination, and fertilisation biology of eggplants in field cultivations, which provided a basis for explaining the effect of a protective biostimulant on these processes. We aimed to investigate the flowering biology of three eggplant hybrids treated with Göemar BM-86®, containing *Ascophylum nodosum* extract, to determine the crucial mechanisms behind the increased flowering and fruit set efficiency and the final effect of increased yield. The flower phenotype (long, medium or short styled), fruit setting, and the number of seeds per fruit were recorded during the two vegetation periods. The numbers of pollen tubes and fertilised ovules in ovaries were evaluated during the generative stage of development to characterise the course of pollination and fertilisation for all types of flowers depending on the cultivar and biostimulant treatment. The expression of heterostyly depended on the eggplant genotype, age of the plant, fruit load, and biostimulant treatment. Domination by long-styled flowers was observed, amounting to 41%, 42%, and 55% of all flowers of "Epic" $F_1$, "Flavine" $F_1$, and "Gascona" $F_1$, respectively. This flower phenotype contained the highest number of pollen tubes in the style and the highest number of fertilised ovules. The biostimulant had a positive effect on the flower and fruit set numbers, as well as on the pollination efficiency in all genotypes. *Ascophylum nodosum* extract could be used as an efficient stimulator of flowering and fruit setting for eggplant hybrids in field conditions in a temperate climatic zone.

**Keywords:** *Ascophyllum nodosum*; *Solanum melongena*; heterostyly; pollination efficiency

## 1. Introduction

Eggplant is a photoperiodically inert plant with bisexual and partially self-pollinating flowers, although cross-pollination increases the effectiveness of fruit setting [1].

The downward-facing flowers are born solitary or in clusters. The eggplant produces three types of flowers: With a long-style pistil, where the stigma is localised above the anthers; with a medium-style pistil, where the stigma is at the same level as the anthers; and with a short-style pistil, where the stigma is below the anthers (Figure 1). This flower character promotes outcrossing between morphs via delivery and uptake of pollen by pollinators [2,3]. The stamen pores of the long- and medium-styled pistils are localised above or close to the stigma, favouring self-pollination. On the contrary, the stigmas of short-styled pistils are inside the downward-facing anther cone, making self-pollination difficult [4–6].

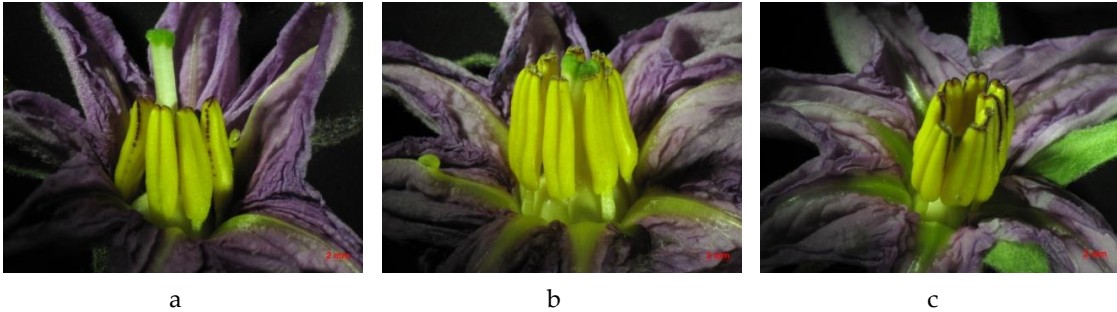

| a | b | c |

**Figure 1.** Stylar heteromorphism in eggplant: The flower with long-styled pistil of "Flavine" $F_1$ (**a**), medium-styled pistil of "Gascona" $F_1$ (**b**), short-styled pistil of "Epic" $F_1$ (**c**).

Anthers are ready to release pollen and the stigma is receptive from the first opening of the flower (Figure 2). Stigma receptiveness gradually decreases with the plant's age, and by the fifth day of flowering, receptiveness is negligible, and the stigma turns brown [2].

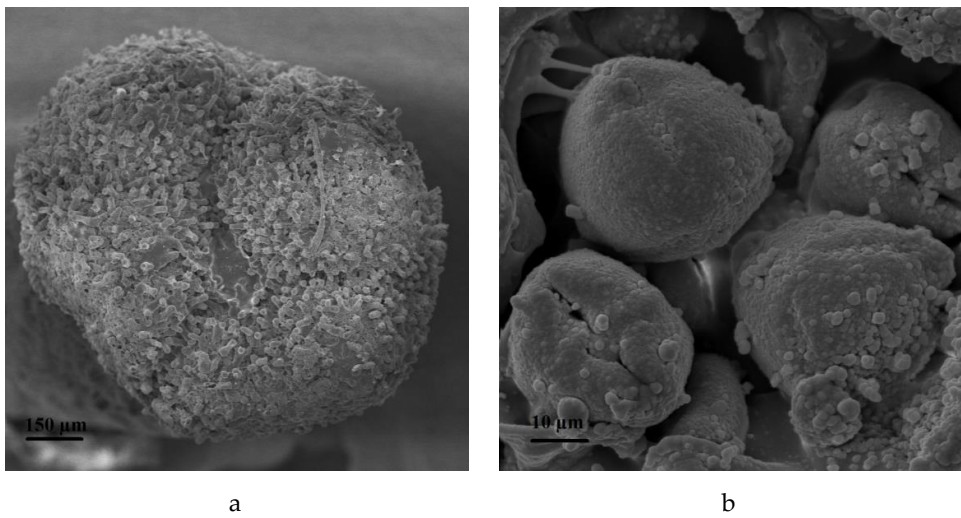

| a | b |

**Figure 2.** Stigma in eggplant pistil with visible papillae in a receptive phase, (**a**), and pollen grains (**b**) in "Epic" $F_1$ by scanning electron microscopy.

All types of flowers are found in the same plant and even within the same cluster. The expression of heterostyly depends on the plant genotype, the age of the plant, fruit load, environmental conditions, and growing practices. Generally, domination by long-styled flowers has been reported, amounting to 50–100% of all flowers [7–9]. The higher fruit setting efficiency of this phenotype results from well-developed nodules with high pollen absorption capacity. However, the development of the ovules and their position in the placenta, as well as pollen grain shape, size, and amount in anthers, were nondifferentiated among long-, medium-, and short-styled pistils [6,10], although Wang et al. [11] demonstrated that lower fruit setting from short-styled flowers resulted from stigma-pollen incompatibility. The bumblebee (*Bombus terrestris*) is the most effective eggplant pollinator for plants under covers. Yield increase and better fruit quality are considered to be the major benefits of bumblebee application as compared to self-pollination or inflorescence vibrating [8,12]. Optimisation of eggplant yield in unfavourable conditions could also be achieved by introducing the cultivation of parthenocarpic cultivars [13]. Pollination leading to fruit and seed formation is associated with the production of endogenous growth regulators such as auxins. In this respect, the use of fruit-setting using auxin-based growth regulators has also been recommended to enhance fruit setting under suboptimal temperatures [5,14]. Investigations on the control of eggplant flowering through growth regulators have been successively performed since the end of the 20th century, but their results have been inconclusive [15,16]. Eggplant tolerance to biotic and abiotic stresses can be

managed through grafting. The effects of rootstock/scion combinations on eggplant performance were investigated in terms of yield and fruit quality [17,18]. It can be assumed that this technique affects the flowering biology as well, but this issue needs future investigation. In Poland, eggplants are cultivated mainly under unheated foil covers from spring to autumn. To lower costs, cultivation is also performed in open fields where air temperatures may fall below the optimum, causing a reduction in flowering and fruit setting [19]. A promising way to control eggplant generative development could be biostimulant application. Biostimulants have been a focus of global interest of the scientific community since the end of the 20th century, giving promising results in different branches of agriculture as stimulators of crops growth, stress tolerance and yield [20,21]. Seaweed extracts (SWE) are among the main biostimulants, recognised as nontoxic, nonpolluting and nonhazardous to various organisms [22,23]. The majority of the SWE formulations are based on the extract of the brown algae *Ascophyllum nodosum* (L.) Le Jolis. Although seaweed extracts are heterogeneous in nature, the leading companies standardise their chemical composition to ensure consistent product quality [24,25]. Some authors have reported the stimulatory effect of seaweed extracts on eggplant yield [26,27], but there are no references on the flowering biology of this species as affected by SWE biostimulation. SWE action is extremely complex, but interdisciplinary investigation of biostimulant vs plant interactions may shed new light on the effective utilisation of these promising bioproducts in horticulture.

We hypothesise that seaweed extract affects the flowering and fruit setting of eggplant in a multidirectional manner. The reaction of plants to biostimulant treatment depends on the flowering biology of the cultivars, particularly the proportions of different flower phenotypes and their fertility. We aimed to investigate the flowering biology of three eggplant hybrids treated with seaweed extract Göemar BM-86® (Arysta LifeScience North America, LLC) to determine the crucial mechanisms behind the final effect of increased yield.

## 2. Materials and Methods

### 2.1. Experimental Arrangement

A two-factorial experiment was set up using randomised blocks in three replications, in the years 2013 and 2015, at the University of Agriculture in Krakow, Poland. The investigated eggplant hybrids, "Epic" $F_1$ (Seminis Vegetable Seeds), "Flavine" $F_1$ (Gautier Semences), "Gascona" $F_1$ (Gautier Semences), were selected on the basis of preliminary studies evaluating their performance in field cultivation under temperate climate conditions [26–28], determined by the earliness, vigour, and yield potential of those plants. Biostimulant Göemar BM-86® (Arysta LifeScience North America, LLC) was applied three times in two week intervals as a foliar application, in a dose of 1.5 $dm^3$ $ha^{-1}$. Control plants were sprayed with distilled water. Goemar BM 86® is standardised *Ascophyllum nodosum* (L.) Le Jolis extract, which provides a constant and balanced formulation containing (in %): N, 5.0; Mg, 2.4; S, 3.2, B, 2.07; and Mo, 0.02 [29].

### 2.2. Cultivation Procedures

Eggplant seeds were sown on 1 March 2013 and 3 March 2015 in seed boxes. After three weeks, the seedlings with one fully developed leaf were transplanted into black 40-cell multipots (VEFI, Norway) with a single cell volume of 0.23 $dm^3$. Seedlings were grown in a greenhouse, in temperatures of 20/17 ± 2 °C day/night. The growing medium was peat substrate KlasmanTS2 (Klasmann-Deilmann GmbH, Geeste, Germany). The foliar fertiliser Kristalon Green (Yara, Szczecin, Poland) was applied twice in a dose of 10 g $dm^{-3}$ water during seedling production. A gradual decrease in temperature and irrigation was used for the hardening of seedlings seven days before being transplanted to the experimental field (50°04′ N, 19°51′ E) on 7 May 2013 and 15 May 2015, with spacing of 0.75 × 0.60 m. Experimental plots covered 15 plants per treatment for observations of flowering and fruit setting and an additional 15 plants per treatment for flower collection for microscopic observations. Plots were surrounded by shelterbelts. The soil of the experimental field was Fluvic Cambisol (Humic) according

to the FAO (Food and Agriculture Organization of the United Nations) classification with a $C_{org}$ level of 2% and $pH_{KCl}$ 6.11. Before the field experiment was established, the soil samples were analysed, and doses of fertilisers were applied to achieve a stable content of nutrients (in mg dm$^3$): N, 100; P, 90; K, 220; Ca, 1,100; Mg, 70. Cultivation procedures of weeding, irrigation, and plant protection were performed according to the standard recommendations for eggplant cultivated in field conditions in Poland, described by Sękara [30].

## 2.3. Weather Conditions

The climate of the experimental station is humid continental (Dfb) according to the Köppen's classification. Detailed data concerning the mean air temperature, photosynthetically active radiation (PAR), and the total rainfall during the vegetation seasons in 2013 and 2015 are presented in Table 1. Data were collected from automatic HOBO Pro RH/Temp loggers to assess temperature and a HOBO Weather Station (Onset Comp. Corp., Cape Cod, USA) to assess light characteristics and rainfall at the experimental site. The growing season in 2015 was generally warmer than that in 2013 regarding mean monthly temperatures, with the exception of June. In 2015, precipitation was distributed evenly, while in 2013, 45% of rainfall was recorded in June. A cool September in both years and low PAR caused a continuous decline in eggplant yield (Table 1).

**Table 1.** Mean monthly temperature, photosynthetically active radiation (PAR) and sum of rainfall in vegetation seasons 2013 and 2015.

| Month | 2013 | | | 2015 | | |
|---|---|---|---|---|---|---|
| | Temperature (°C) | PAR ($\mu mol\,m^{-2}\,s^{-1}$) | Sum of Rainfall (mm) | Temperature (°C) | PAR ($\mu mol\,m^{-2}\,s^{-1}$) | Sum of Rainfall (mm) |
| May | 14.3 | 345 | 83 | 13.1 | 357 | 93 |
| June | 17.6 | 392 | 188 | 17.5 | 401 | 40 |
| July | 19.4 | 477 | 28 | 20.4 | 489 | 39 |
| August | 18.8 | 396 | 51 | 21.2 | 357 | 58 |
| September | 12.1 | 256 | 63 | 14.9 | 283 | 60 |

## 2.4. Procedures for Flowering and Fruit Setting Observations

The observations were conducted on 5 plants per replication (15 plants per treatment and cultivar) during the flowering period, from June to September. Single flowers were labeled according to the order of appearance on each plant. The numbers of flowers of particular phenotypes (with long-styled, medium-styled, or short-styled pistil) were recorded after the opening of petals. Then, the number of fruits set from flowers of a particular phenotype was also recorded about one week after fertilisation, when fruit sets reached 1–2 cm in diameter. Flowers which did not set fruits were naturally aborted. Fruits in a stage of harvest maturity were picked to reflect the standard cultivation conditions and to exclude excessive metabolite sink by ripening fruits. Data were calculated and presented as a sum of flowers and fruits per plant per month for the two experimental years separately.

## 2.5. Procedures for Microscopic Observations

At full flowering, 20 pollinated flowers of each phenotype per treatment and cultivar were collected in 2013 and 2015. Data are presented as a sum of observations for investigated seasons, $N = 40$. The styles were isolated and fixed in FAA (formalin-acetic-alcohol), according to Martin's method [31] adapted by Sękara [30]. The germination of pollen on stigmas, growth of pollen tubes, and fertilisation of ovules were examined under fluorescence microscopy with the use of SteREO LUMAR V12 microscope (Carl Zeiss AG, Jena, Germany) (Figure 3). The number of pollen tubes in half of the style and the number of fertilised ovules were evaluated. The numbers of pistils having a number of pollen tubes in the ranges 0–100; 100–200; 200–300; 300–400; 400–500; 500–600; 600–700; 700–800; 800–900; 900–1000 were determined. For fertilised ovules, the following ranges were included: 0–50; 50–100; 100–150; 150–200; 200–250; 250–300; 350–400.

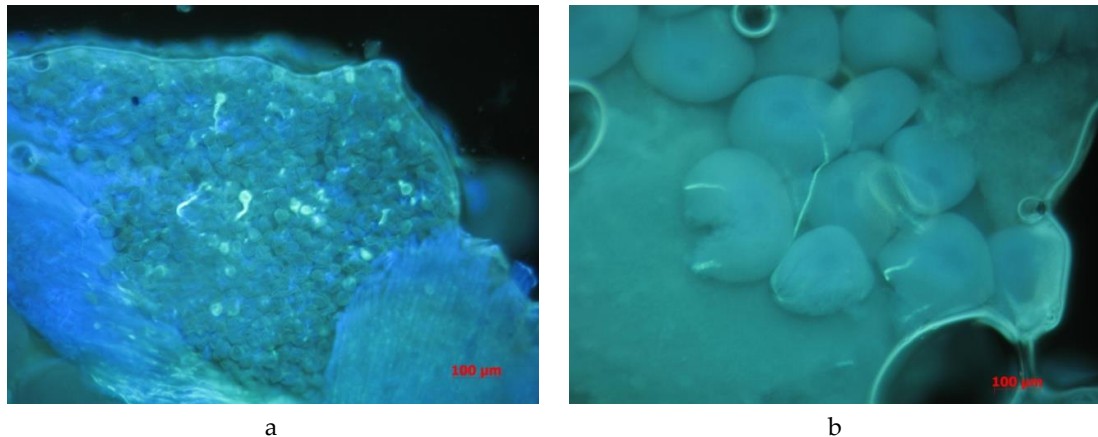

<center>a            b</center>

**Figure 3.** Germination of the pollen on stigmas (**a**) and fertilisation of ovules (**b**) of eggplant observed under fluorescence microscopy after Martin's aniline blue fluorescence technique.

### 2.6. Statistical Analyses

Statistical analyses were performed using the Statistica 12.0 software package (StatSoft Inc., Tulsa, OK, USA). A three-way analysis of variance followed by Tukey's honest significance test was used to determine the main effects of the type of flower, biostimulant, and time of sampling, as well as interactions between main effects, at the $p \leq 0.05$ significance level. Data shown in the tables and figures are averages of three replicates.

## 3. Results

In the conditions of the present experiment, the eggplants started flowering at the beginning of June, while the period of the most intensive flowering fell in August. The investigated hybrids showed flower heterostyly—the presence of long-, medium-, and short-styled flowers was observed for all investigated plants. Moreover, heterostyly expression significantly depended on biostimulant treatment and the age of the plants (Table 2).

**Table 2.** Chosen aspects of flowering and fruit setting of eggplant as depended on fruit type and biostimulant treatment.

| Parameter | Year | Type of Flower | | | | | |
|---|---|---|---|---|---|---|---|
| | | With Long-Styled Pistil | | With Medium-Styled Pistil | | With Short-Styled Pistil | |
| | | C * | B | C | B | C | B |
| | | "Epic" F$_1$ | | | | | |
| Number of particular types of flowers per plant | 2013 | 12.2 d ** | 13.4 d | 5.6 b | 8.2 c | 3.4 a | 3.5 ab |
| | 2015 | 12.1 c | 13.5 c | 6.4 b | 7.6 b | 3.5 a | 3.6 a |
| Number of fruits per plant set from particular types of flowers | 2013 | 2.4 b | 4.2 d | 0.8 a | 2.0 b | 1.0 a | 1.4 ab |
| | 2015 | 3.2 b | 5.6 c | 1.0 a | 3.2 b | 1.0 a | 1.2 a |
| Effectiveness of fruit setting as depended on type of flower (%) | 2013 | 20.7 | 27.6 | 15.6 | 24.7 | 26.7 | 40.0 |
| | 2015 | 33.7 | 48.8 | 18.9 | 46.7 | 26.6 | 36.7 |
| Number of seeds per fruit | 2013 | 332 d | 365 e | 245 b | 289 c | 102 a | 125 a |
| | 2015 | 358 e | 372 f | 255 c | 312 d | 95 a | 138 b |
| | | "Flavine" F$_1$ | | | | | |
| Number of particular types of flowers per plant | 2013 | 7.8 cd | 10.0 d | 6.6 bc | 10.0 d | 4.2 a | 4.8 ab |
| | 2015 | 8.4 b | 11.2 c | 6.8 b | 11.0 c | 3.8 a | 4.8 a |
| Number of fruits per plant set from particular types of flowers | 2013 | 2.0 abc | 2.8 b | 1.4 ab | 2.4 bc | 1.2 a | 1.6 ab |
| | 2015 | 1.8 a | 3.6 b | 1.2 a | 3.2 b | 1.2 a | 2.0 a |
| Effectiveness of fruit setting as depended on type of flower (%) | 2013 | 26.9 | 28.7 | 22.8 | 26.0 | 20.0 | 33.3 |
| | 2015 | 26.0 | 36.2 | 16.7 | 27.2 | 30.0 | 41.1 |
| Number of seeds per fruit | 2013 | 312 c | 328 c | 258 b | 325 c | 142 a | 155 a |
| | 2015 | 322 c | 333 d | 289 b | 316 c | 134 a | 143 a |

**Table 2.** *Cont.*

| Parameter | Year | Type of Flower | | | | | |
|---|---|---|---|---|---|---|---|
| | | With Long-Styled Pistil | | With Medium-Styled Pistil | | With Short-Styled Pistil | |
| | | C * | B | C | B | C | B |
| | | "Gascona" $F_1$ | | | | | |
| Number of particular types of flowers per plant | 2013 | 7.6 c | 8.0 c | 5.6 b | 8.2 c | 3.4 a | 3.6 a |
| | 2015 | 7.6 bc | 7.8 c | 6.0 b | 7.8 c | 4.2 a | 3.8 a |
| Number of fruits per plant set from particular types of flowers | 2013 | 2.0 abc | 3.2 c | 1.6 ab | 2.4 bc | 1.0 a | 0.8 a |
| | 2015 | 2.4 ab | 3.0 b | 1.6 a | 2.2 ab | 1.4 a | 1.4 a |
| Effectiveness of fruit setting as depended on type of flower (%) | 2013 | 26.2 | 41.5 | 22.6 | 29.1 | 23.3 | 20.0 |
| | 2015 | 38.7 | 37.1 | 11.7 | 30.6 | 36.7 | 33.3 |
| Number of seeds per fruit | 2013 | 289 d | 322 e | 257 c | 285 d | 78 a | 127 b |
| | 2015 | 321 e | 356 f | 269 c | 297 d | 98 a | 159 b |

\* C, control; B, biostimulant; \*\* Means within rows, followed by different letters, are significantly different at $p \leq 0.05$, $N = 3$. Comparisons were performed with the use of Tukey's honest significance test.

Among 23 flowers set by "Epic" $F_1$ plant during one vegetation period, 55% had long-style pistils, 30% had medium-style pistils, and 15% had short-style pistils. "Epic" $F_1$ plants produced 7 fruits during the vegetation season, on average; 57% of these were from long-styled flowers, 26% from medium-styled flowers, and 17% from short-styled flowers. Biostimulant treatment significantly increased the number of only medium-styled flowers in 2013 and the number of fruits set by long- and medium-styled flowers in both vegetation periods. The most effective in fruit setting were long-styled flowers. The biostimulant positively affected the percentage of fruits set by all flower phenotypes and the number of seeds, with the exception of flowers with short-styled pistils in 2013. The first fruits were collected at the end of July. The highest number of long- and medium-styled flowers was observed in August; the lowest was observed in September (Figure 4, Table 3).

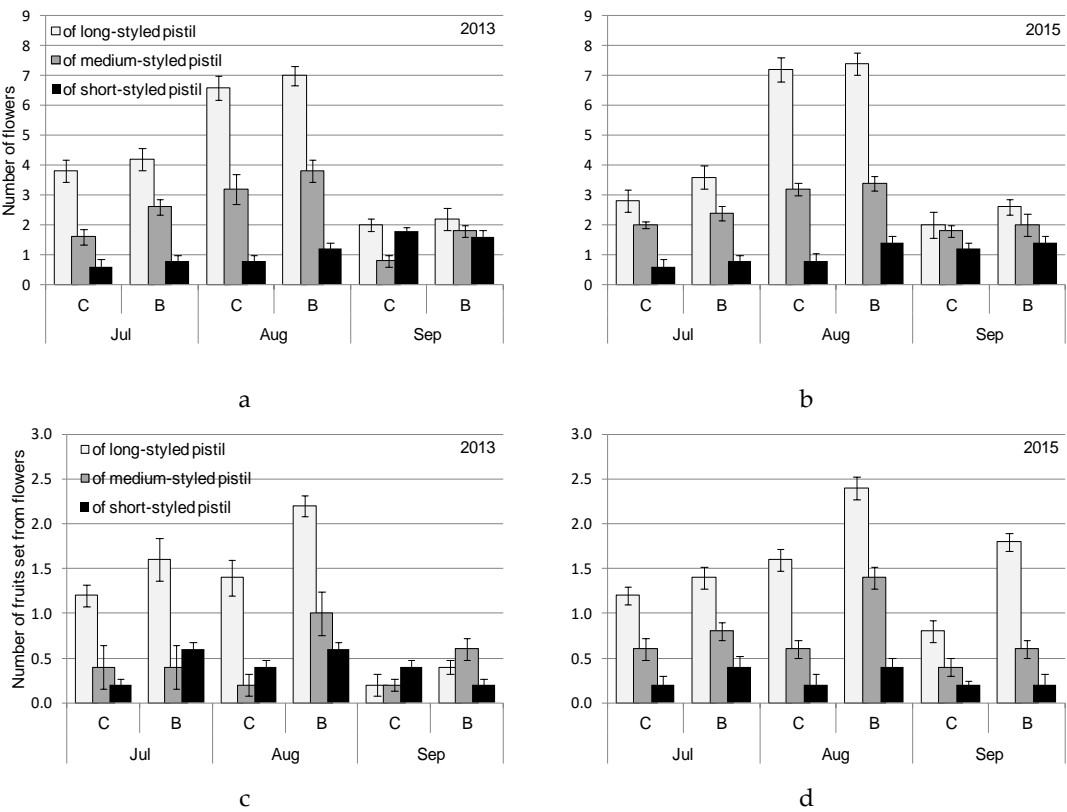

**Figure 4.** The course of flowering and fruit setting of "Epic" $F_1$ eggplant as depended on fruit type and biostimulant treatment. C, control; B, biostimulant. Bars represent mean number of flowers per plant in 2013 (**a**), 2015 (**b**) and fruits per plant in 2013 (**c**), and 2015 (**d**) (error bars indicate SE).

**Table 3.** Results of ANOVA for parameters of flowering and fruit setting of "Epic" F$_1$ eggplant presented in Figure 4.

| ANOVA Source of Variation | "Epic" F$_1$ | | | |
|---|---|---|---|---|
| | No of Flowers 2013 | No of Fruits 2013 | No of Flowers 2015 | No of Fruits 2015 |
| Type of flower (F) | *** | *** | *** | *** |
| Biostimulant (B) | ** | * | *** | *** |
| Month (M) | *** | *** | * | ** |
| F × B | ns | * | ns | * |
| F × M | *** | *** | *** | ns |
| B × M | ns | ns | ns | * |

Levels of significance for ANOVA: * $p \leq 0.05$; ** $p \leq 0.01$; *** $p \leq 0.001$; ns, not significant; $N = 3$. Comparisons were performed with the use of Tukey's honest significance test.

The number of short-styled flowers increased in line with aging of the plants. The number of fruits set from long- and medium-styled flowers increased from July to August, then decreased in September. We observed, on average, 106 pollen tubes in the short-styled pistils, 422 in medium-styled pistils, and 610 in long-styled pistils collected from control plants and 129, 490, and 778 pollen tubes, respectively, collected from biostimulant-treated plants (Figure 5). The ovaries of the short-styled flowers contained approximately 36 fertilised ovules, and more fertilised ovules were found in the remaining types of flowers: 199 and 225 in medium- and long-styled flowers, respectively, produced by control plants. The flowers of biostimulant-treated plants contained 39%, 32%, and 36% more fertilised ovules in the ovaries of short-, medium-, and long-styled flowers, respectively.

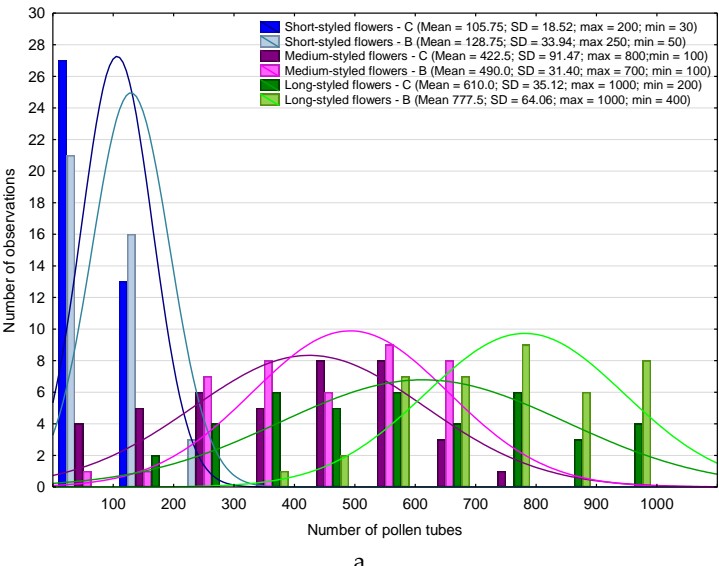

a

**Figure 5.** *Cont.*

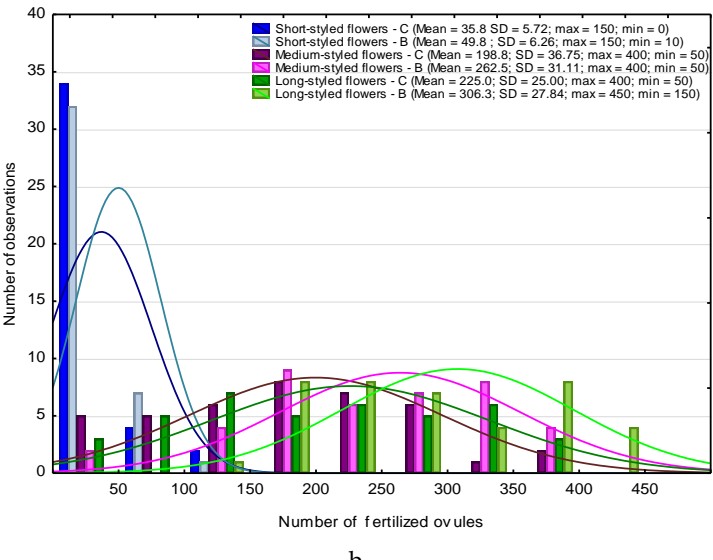

b

**Figure 5.** Number of pollen tubes (**a**) and fertilised ovules (**b**) in the styles of different flower types in "Epic" F$_1$.

The "Flavine" F$_1$ plants produced 22 flowers during the vegetation period; 41% had long-styled pistils, 38% had medium-styled pistils, and 19% had short-styled ones (Table 2). The number of fruits collected from a plant was 6 on average; 42% of these were from long-styled flowers, 34% from medium-styled flowers, and 24% from short-styled flowers. Biostimulant treatment significantly increased the number of long-styled flowers in 2015, the numbers of medium-styled flowers in 2013 and 2015, and the numbers of fruits set by long- and medium-styled flowers in 2015. The most effective in fruit setting were long-styled flowers. The biostimulant positively affected the percentage of fruits set by all flower phenotypes and the number of seeds in fruits born by long-styled flowers in 2015 and by medium-styled flowers in both years of the experiment. The highest number of long- and medium-styled flowers was observed in August; the lowest was in September (Figure 6, Table 4). The number of fruits set from long-styled flowers was the highest in August. We observed, on average, 149 pollen tubes in the middle of the style in the short-styled flowers of control plants, 410 in medium-styled flowers, and 595 in long-styled ones (Figure 7). In biostimulant-treated flowers, a 23% higher number of pollen tubes was observed in short-styled pistils, 16% higher in medium-styled pistils, and 20% higher in long-styled pistils. The ovaries of the short-styled flowers collected from the control plants contained, on average, 54 fertilised ovules; more fertilised ovules were found in the remaining types of flowers: 208 and 243 in medium- and long-styled flowers, respectively. The numbers of fertilised ovules in analogous types of flowers collected from biostimulant-treated plants were 102%, 24%, and 23% higher, respectively.

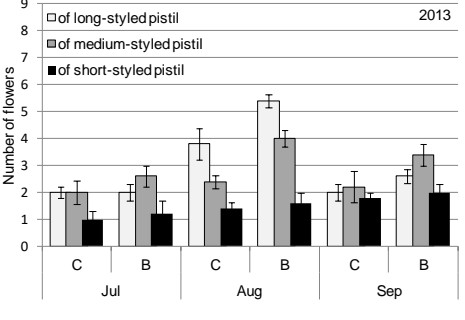

a

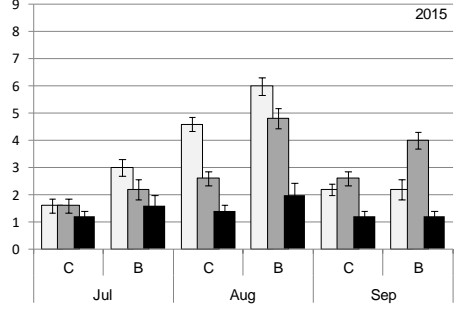

b

**Figure 6.** *Cont.*

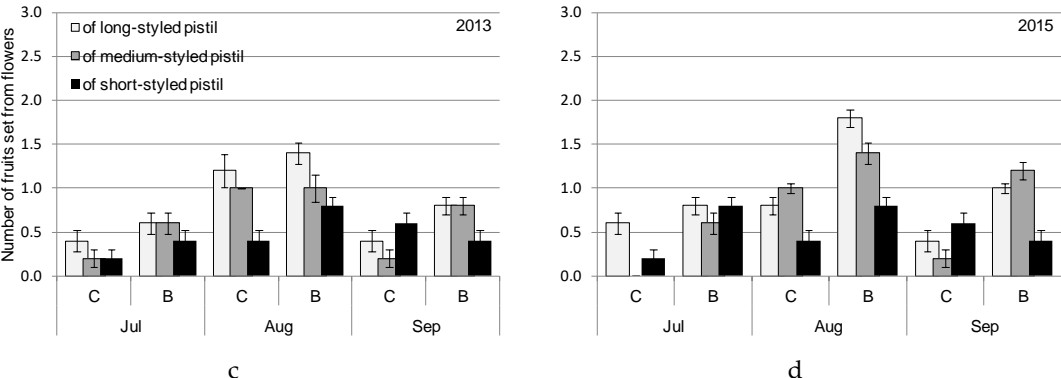

c            d

**Figure 6.** The course of flowering and fruit setting of "Flavine" $F_1$ eggplant as depended on fruit type and biostimulant treatment. C, control; B, biostimulant. Bars represent mean number of flowers per plant in 2013 (**a**), 2015 (**b**) and fruits per plant in 2013 (**c**), and 2015 (**d**) (error bars indicate SE).

**Table 4.** Results of ANOVA for parameters of flowering and fruit setting of "Flavine" $F_1$ eggplant presented in Figure 6.

| ANOVASource of Variation | "Flavine" $F_1$ | | | |
|---|---|---|---|---|
| | No of Flowers 2013 | No of Fruits 2013 | No of Flowers 2015 | No of Fruits 2015 |
| Type of flower (F) | *** | *** | *** | ** |
| Biostimulant (B) | ** | *** | *** | *** |
| Month (M) | *** | *** | *** | *** |
| F × B | ns | * | * | ns |
| F × M | *** | *** | *** | ns |
| B × M | ns | * | * | ns |

Levels of significance for ANOVA: * $p \leq 0.05$; ** $p \leq 0.01$; *** $p \leq 0.001$; ns, not significant; $N = 3$. Comparisons were performed with the use of Tukey's honest significance test.

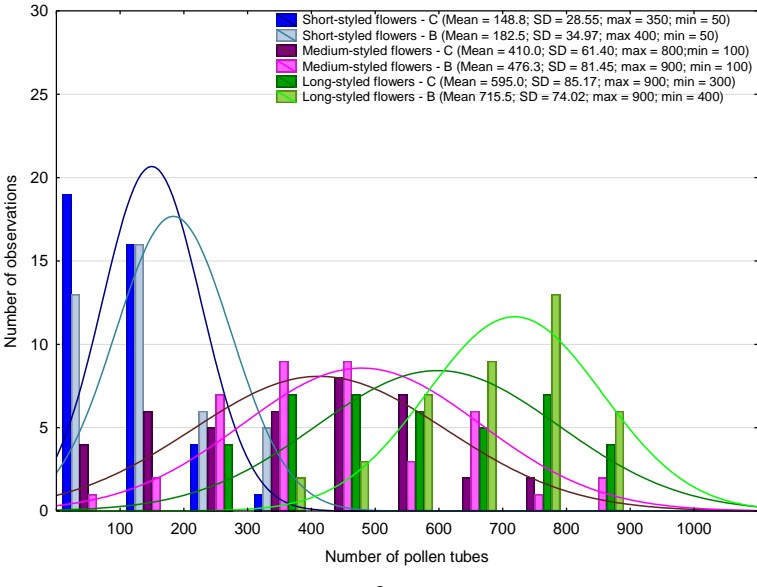

a

**Figure 7.** *Cont.*

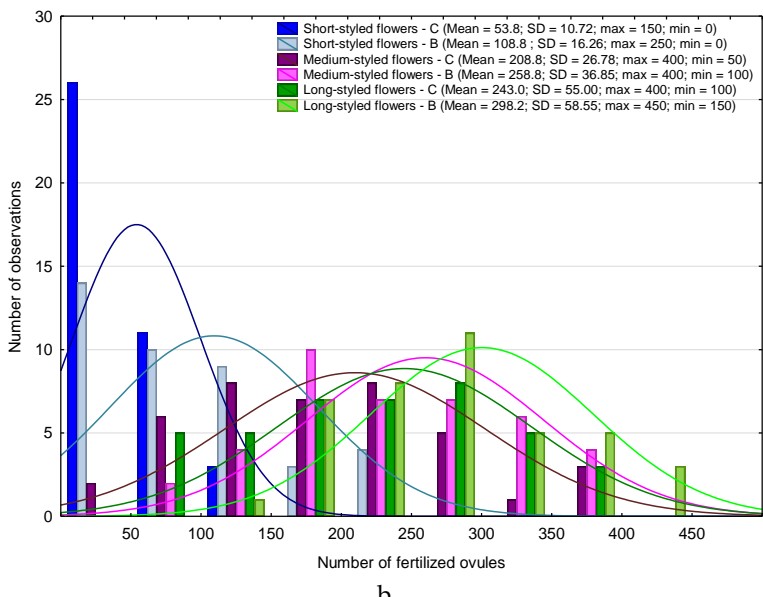

**Figure 7.** Number of pollen tubes (**a**) and fertilised ovules (**b**) in the styles of different flower types in "Flavine" $F_1$.

"Gascona" $F_1$ plants produced 18 flowers during the vegetation period, with 42% of these being long styled, 38% medium styled, and 20% short styled (Table 2). The number of fruits collected from a plant was 6, on average; 46% of these were from long-styled flowers, 34% from medium-styled flowers, and 20% from short-styled flowers. Biostimulant treatment significantly increased the number of medium-styled flowers in 2013 and 2015 but did not affect the number of fruits. The most effective in fruit setting were long-styled flowers. The biostimulant positively affected the percentage of fruits set by long-styled flowers in 2015 and medium-styled flowers in both years, but it negatively affected the effectiveness of fruit setting by short-styled flowers. Biostimulant treatment positively affected the seed number (Table 2). The highest number of long-, and medium-styled flowers was observed in August; the lowest was in September (Figure 8, Table 5). The number of fruits set from long- and medium-styled flowers increased from July to August, then decreased in September. Long-, medium-, and short-styled flowers were analysed regarding the number of pollen tubes in the styles. Differences in the course of pollination and fertilisation between investigated cultivars concerned the number of pollen tubes and fertilised ovules in the pistils. For control "Gascona" $F_1$ plants, we observed, on average, 119 pollen tubes in the middle of the style in the short-styled flowers, 418 in medium-styled flowers, and 595 in long-styled ones (Figure 9). The ovaries of the short-styled flowers contained approximately 0–50 fertilised ovules, while more fertilised ovules were found in the remaining types of flowers: 200–400 in medium- and long-styled flowers. The flowers of control plants contained lower numbers of both pollen tubes and fertilised ovules in all types of flowers.

**Table 5.** Results of ANOVA for parameters of flowering and fruit setting of "Gascona" $F_1$ eggplant presented in Figure 8.

| ANOVASource of Variation | "Gascona" $F_1$ | | | |
|---|---|---|---|---|
| | No of Flowers 2013 | No of Fruits 2013 | No of Flowers 2015 | No of Fruits 2015 |
| Type of flower (F) | *** | *** | *** | * |
| Biostimulant (B) | * | *** | ns | ns |
| Month (M) | *** | ns | *** | *** |
| F × B | * | ns | * | ns |
| F × M | *** | ns | *** | * |
| B × M | ns | ns | ns | ns |

Levels of significance for ANOVA: * $p \leq 0.05$; *** $p \leq 0.001$; ns, not significant; $N = 3$. Comparisons were performed with the use of Tukey's honest significance test.

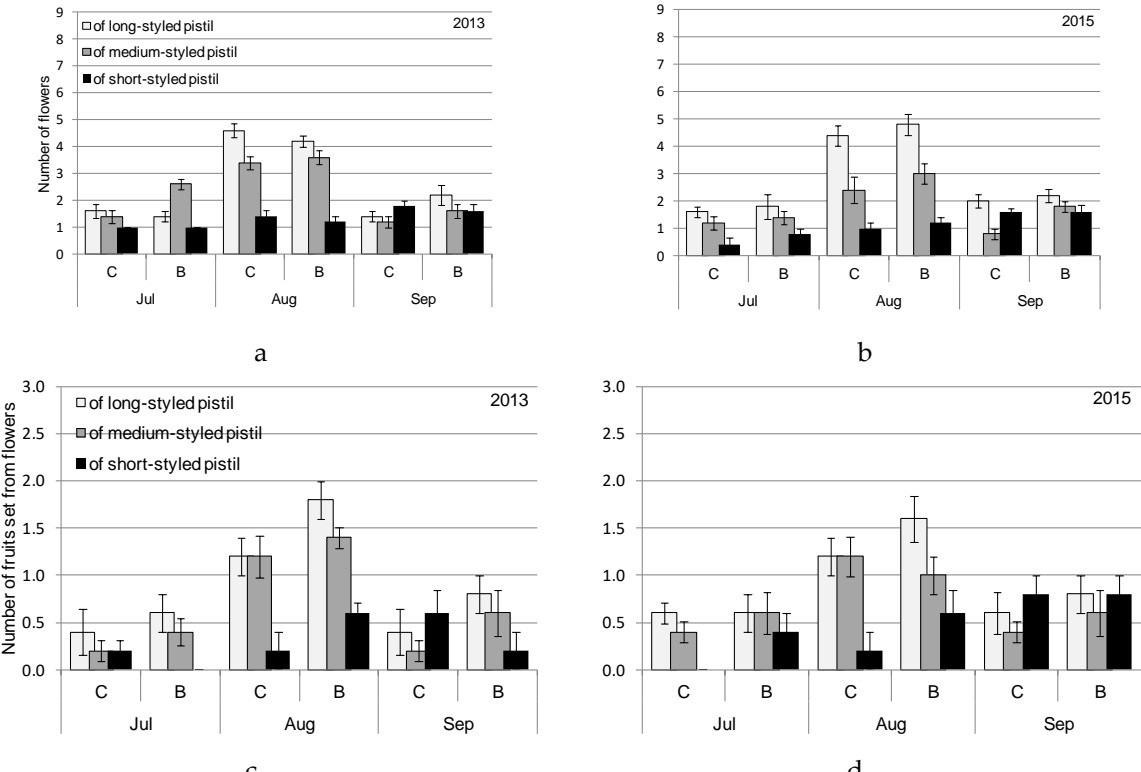

**Figure 8.** The course of flowering and fruit setting of "Gascona" $F_1$ eggplant as depended on fruit type and biostimulant treatment. C, control; B, biostimulant. Bars represent mean number of flowers per plant in 2013 (**a**), 2015 (**b**) and fruits per plant in 2013 (**c**), and 2015 (**d**) (error bars indicate SE).

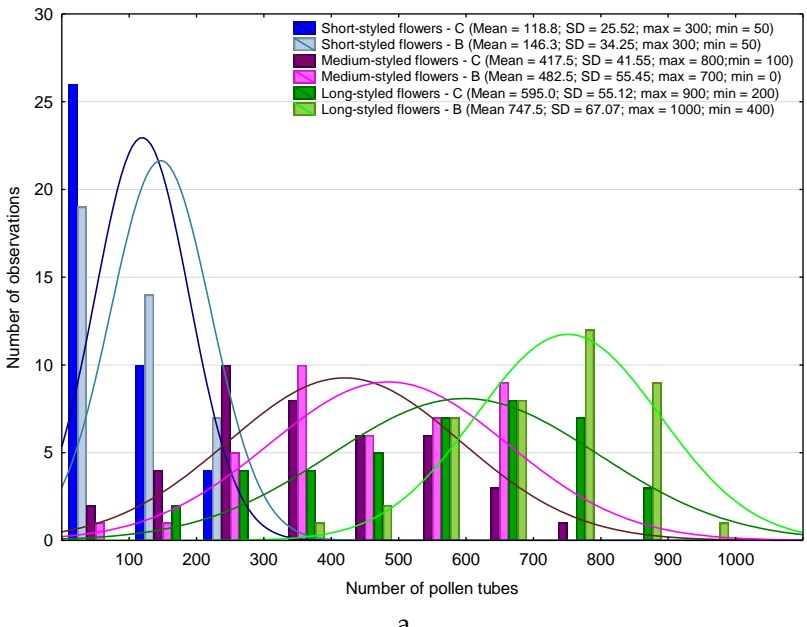

**Figure 9.** *Cont*.

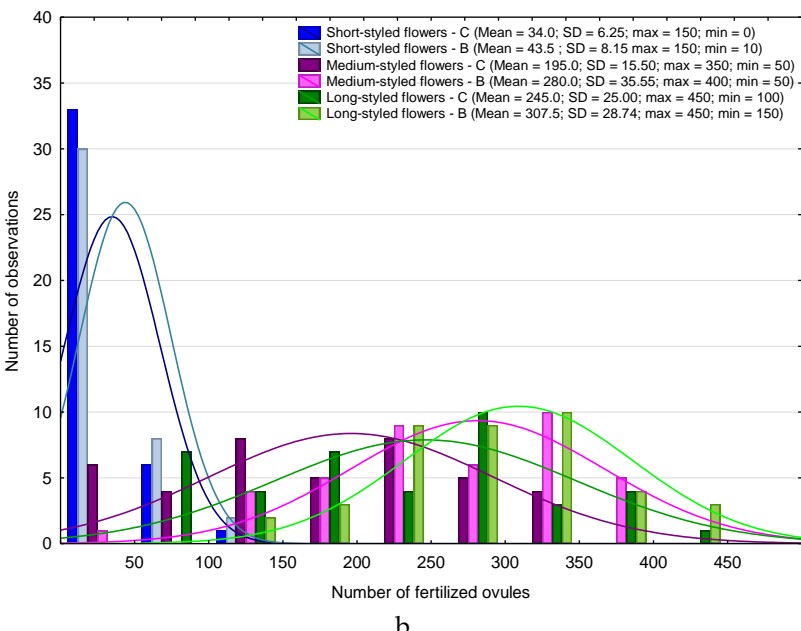

b

**Figure 9.** Number of pollen tubes (**a**) and fertilised ovules (**b**) in the styles of different flower types in "Gascona" $F_1$.

## 4. Discussion

### 4.1. Heterostyly Expression in Eggplant as Affected by Biostimulant Treatment and Cultivar

The recent research aimed to develop an overview of the heterostyly phenomenon in eggplant, and its implications on fruit setting biology. We demonstrated the presence of three phenotypes of flowers and the differentiated fertility of them, specific to the investigated hybrids. Generally, long-styled flowers dominated, but the fruit setting efficiency was not directly determined by the flower phenotype. A study by Srinivas et al. [32] indicated that for two eggplant hybrids of Indian breeding, with long, green and round, purple fruits, 80% of fruits were set by long-styled flowers, whereas 20% of the fruits were set by medium-styled flowers and no fruit by short-styled flowers. The partial sterility of short-styled flowers demonstrated in the cited research was due to small stigmas with under-developed papillae on which pollen grains failed to germinate. The short-styled flowers of the eggplant hybrids which are the subject of the present investigations were fertile, although the lowest number of pollen tubes and fertilised ovules was observed in this flower phenotype. Despite this fact, "Epic" $F_1$ and "Gascona" $F_1$ plants set about 20% of fruits from short-styled flowers. For "Flavine" $F_1$, the percentage of fruits set by this mentioned flower phenotype was 30%. Sękara and Bieniasz [6] determined that the ovules of short-styled pistils were typically developed, but that their fruit setting efficiency was low. On the contrary, results by Hazra et al. [33] indicated full sterility of short-styled flowers due to some problem related to ovary development. Observations with the use of a fluorescence microscope allowed us to verify the correct growth of pollen tubes in the styles of all types of pistils but their number was significantly affected by flower type and biostimulant treatment and by cultivar to a lesser extent. This observation is contrary to the results of Wang et al. [11], who determined that the structure of the stigmatic surface in short-styled flowers inhibited pollen germination. On the grounds of highly genotype-dependent heterostyly expression in eggplant, results on short-styled pistil performance may be divergent.

Application of Göemar BM-86® caused an increase in the numbers of pollen tubes and fertilised ovules. This phenomenon was common for all types of flowers and is directly attributable to pistil characteristics. Biostimulant-treated and control plants were not isolated, so they both could act as pollen donors. The effect of biostimulants on pollen production and fertility should also be an object of future research. Based on the available literature, we can only conclude that a wide pool of

bioactive seaweed extract compounds provided balanced development and enhanced the flowering and fruiting of the investigated eggplant hybrids. The biostimulant-treated plants could be able to develop a better canopy for effective interception of light and—through a significant reduction in interplant competition for solar energy and nutrients—build suitable carbohydrate reserves earlier. Such mechanisms beyond increased flowering and fruit setting in seaweed extract treated plants were proposed by Arthur et al. [34] for bell peppers and by Helaly et al. [35] for tomatos.

The increasing number of short-styled flowers in line with plant aging, in the conditions of the present experiment, could be the result of increasing fruit load with the vegetation season's flow. Having well-developed anthers, short-styled flowers act as pollen donors to provide reproductive success. Araméndiz Tatis et al. [9] demonstrated that short-styled flowers of the "Lilac" eggplant landrace and "Long Purple" increased male fitness and thus produced an imbalance in functioning between male and hermaphrodite flowers. According to Khah et al. [36], fruit load negatively affected style length but not anther cone length in eggplant, even under favourable climatic conditions. The investigated hybrids could reduce energy outlines by creating flowers with reduced pistil and decreased fertility at the end of the vegetation period, but do so while producing pollen in the normally shaped anthers, promoting male behaviour. Short-styled flowers could be borne by fruit-loaded plants as a source of pollen for insects. The construction of the eggplant stamens is an expression of adaptation to pollination through vibrations. Such an adaptation limits the potential pollinators to species that are able to introduce vibrations into anthers, including bumblebees [13]. Bumblebees commonly visited eggplant flowers in the conditions of the presented experiment.

## 4.2. Biostimulant-Affected Flower and Fruit Set Effectiveness

Biostimulants have shown promising results in promoting flowering and reducing the fruit drop agents in many fruit trees, like apple, avocado, clementine, orange, olive, and pomegranate [37–40]. In this respect, seaweed extracts enriched in microelements are the most effective [41]. Vegetables with edible fruits are characterised by competition between the flowers and fruits at different stages of growth and in different positions in relation to inflowing assimilates [42]. Dropping of flowers, typical for eggplant, could be the mixed effect of lack of pollination or limited inflow of assimilates and the phenomenon of domination of fruit producing growth regulators. In the present research, hybrids treated with seaweed extract bore more flowers and fruits than did untreated ones. More intensive flower setting was elicited either by improved plant growth through seaweed extract application or by endogenous components, especially cytokinins, which enhance nutrient partitioning in vegetative plant organs and increase in the transport of assimilates to the growing fruits. A similar effect was observed for eggplant treated with seaweed extract by Abd El-Gawad and Osman [43]. Under the conditions of the presented experiment, biostimulant application also increased the number of pollen tubes and fertilised ovules in all types of flowers of the investigated cultivars. The overall positive influence of seaweed extracts on the plants resulted in better reproductive effectiveness and increased fruit yield and quality, described in detail by Pohl et al. [26,27]. Gómez-Cadenas et al. [44] investigated the effect of a biostimulant product containing macronutrients on citrus fruit set abscission. The beneficial effects of the biostimulant resulted from an increase in the photosynthetic efficiency which led to better transport of carbohydrates from leaves to fruit sets. Seaweed-treated apple trees also showed higher leaf chlorophyll contents and increased rates of photosynthesis and respiration due to treatment decreasing the oscillations in yield between "on" and "off" years and increasing the average fruit weight on plants affected by too high a crop load [45]. Pollination and fertilisation are very stress-sensitive stages of development [1]. Based on research on tomato, low temperatures, especially during the night, are not detrimental to ovule development but could affect stigma and style function [5]. Pollen viability is the highest at 20–22 °C [14], while the mean temperatures for the flowering period (July–September) were 16.8 and 18.8 °C in 2013 and 2015, respectively. Pollen development and viability depend on carbohydrate supply [5], so the increased photosynthetic performance of biostimulant-treated plants could improve sugar partitioning to developing pollen grains. The bioactive compounds of seaweed

extracts enhance the tolerance of eggplant to abiotic stresses [46] and this tolerance can also cover the generative reproduction of this crop in temperate regions.

## 5. Conclusions

Eggplant is a warm climate crop and is cultivated for fruits, widely used in many world cuisines because of their unique taste and dietetic value. Nonoptimal growing conditions, especially in temperate climatic zones, affect plant flowering and fruit setting. Biostimulant application in the experiments presented herein affected the flowering biology of eggplant cultivars in different ways. Generally, the biostimulant positively affected the percentages of the most fertile medium- and long-styled flowers and the effectiveness of fruit setting by all flower phenotypes. Increased numbers of pollen tubes and fertilised ovules in all types of flowers of the investigated cultivars were noted. The overall positive influence of Göemar BM-86® on the plants resulted in increased reproductive effectiveness. Biostimulant application seems to be a promising solution to improve eggplant flowering and fruit setting in unfavourable growing conditions.

**Author Contributions:** Conceptualization, A.P. and A.S.; methodology, A.P. and A.S.; software, A.P. and A.S.; validation, A.P. and A.S. and A.K.; formal analysis, A.P.; investigation, A.P. and A.G.; resources, A.P. and A.S.; data curation, A.P. and A.S.; writing—original draft preparation, A.P. and A.S.; writing—review and editing, A.P. and A.S. and A.K.; visualization, A.P. and A.S.; supervision, A.S.

**Funding:** This research was funded by the Ministry of Science and Higher Education of the Republic of Poland.

**Conflicts of Interest:** The authors declare no conflict of interest.

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
