# Peer review of "Biostimulant Application Enhances Fruit Setting in Eggplant—An Insight into the Biology of Flowering"

_agronomy, doi:10.3390/agronomy9090482_

Round 1
Reviewer 1 Report
I have made some comments on the manuscript. I have not edited throughout so I recommend extensive editing for language and style.
Not necessarily for this study but please review the comment on the need for the use of positive controls to determine biostimulatory efficacy (vs fertilser effect of micro and macronutrients).?
The introduction is too long, in my opinion, and can be shortened considerably. I read the methods and suggested use of a positive control to be considered in FUTURE studies. This study does not need a positive control, (though interpretation might have been better if one were used). The study is well done and the results are interesting for extending the range of cultivation of eggplants in temperate regions. It should be published, but the manuscript as submitted requires major revision.Author Response
Dear Reviewer,
Thank you very much for the constructive comments and overall opinion on the manuscript. All your suggestions were taken in to account in the revised version, namely:
1. The text of the revised manuscript is after professional English proofreading by MDPI English Editing Service.
2. Thank you for this advice, we will use positive control in our future studies on biostimulants.
3. We shortened the introduction, although another Reviewer recommended supplementing this chapter with additional information... I hope the final result is satisfactory.
With kind regards,
Agnieszka Sękara
Reviewer 2 Report
The article submitted by the authors to Agronomy aims to investigate flowering biology of three eggplant hybrids treated with seaweed extract to point the crucial mechanisms behind the final effect in an increased yield.
In this work, the authors looked at some aspects such as, number of particular types of flowers per plant, number of fruits per plant set from particular types of flowers, effectiveness of fruit setting as depended on type of flower (%), number of seeds per fruit, number of flowers and number of fruits set from flowers. Furthermore, number of pollen tubes and number of fertilized ovules in relation to the number of observations were also determined.
Overall, the "Introduction" section is well organized, however, some information about other techniques (e.g. vegetable grafting) that can affect yield in eggplant must be mentioned. In this regards, please cite the following references remarking the importance of the cultivar/scion used:
- Sabatino, L., Iapichino, G., D'Anna, F., Palazzolo, E., Mennella, G., Rotino, G.L. 2018. Hybrids and allied species as potential rootstocks for eggplant: Effect of grafting on vigour, yield and overall fruit quality traits. Scientia Horticulturae, 228, 81-90.
- Sabatino, L., Iapichino, G., Rotino, G.L., Palazzolo, E., Mennella, G., D'Anna, F. 2019. Solanum aethiopicum gr. gilo and Its Interspecific Hybrid with S. melongena as Alternative Rootstocks for Eggplant: Effects on Vigor, Yield, and Fruit Physicochemical Properties of Cultivar 'Scarlatti'.
The experimental design was properly performed and the statistical analysis sounds. However, the significant interactions reported in Table 3 were not studied. Consequently, please add a Table that describe the statistical significant interactions between the fix factors (type of flower, biostimulant and month). The discussion section is well organized. From my point of view the paper is of high originality, well written and organized, with high significance to the field as well as of high interest for the Agronomy readers.
To summarize, I believe that the results of this study are very interesting and unique and the paper is well written with a good discussion.
Author Response
Dear Reviewer,
Thank you very much for the constructive comments and overall opinion on the manuscript. All your suggestions were taken in to account in the revised version, namely:
With kind regards,
Agnieszka Sękara
Reviewer 3 Report
This manuscript concludes that Biostimulant Application ( in their case they have used Göemar BM-86®, containing Ascophylum nodosum) Enhances Fruit Setting in Eggplant, which is a piece of common knowledge.
Although results are encouraging for the early stages, I think this experiment should be used to determine the effect of this Biostimulant Application on yield, fruit quality parameters, fruit size, etc. Similarly, for other important economically important biochemical traits for eggplant.
For the current version of this manuscript, I would like to suggest the authors improve the introduction, and discussion sections as these sections are hard to follow.
Author Response

(The authors gave the same response as above.)

Reviewer 4 Report
Revision is required.

Author Response
Dear Reviewer,
Thank you very much for the constructive comments and overall opinion on the manuscript. Your suggestions, provided in pdf file were taken in to account in the revised version.
The two comments we decided to remain unchanged are:
Answer: Generally, investigations with the effect of biostimulants on flowering and fruit setting efficiency were started on fruit crops, that’s why we mentioned this species. Vegetable references are mentioned in the following part of this paragraph.
With kind regards,
Agnieszka Sękara
Reviewer 5 Report
The manuscript is well prepared. However, please check carefully for spelling mistakes (e.g. line 11: "September" is correct, line 279 "pollen" etc.). In the Materials and Methods section, sometimes the company details are missing (e.g. line 113, line 154).
Table or figure legends are without the number of replicates and details on the statistical analyses (name of test used etc.). Please add them. It is not sufficient to mention them only in the main text.
Line 162: Delete "the use"
Author Response

(The authors gave the same response as above.)

Round 2
Reviewer 3 Report
Thank you for the changes you have made.